# The Target Therapy Hyperbole: “KRAS (p.G12C)”—The Simplification of a Complex Biological Problem

**DOI:** 10.3390/cancers16132389

**Published:** 2024-06-28

**Authors:** Massimiliano Chetta, Anna Basile, Marina Tarsitano, Maria Rivieccio, Maria Oro, Nazzareno Capitanio, Nenad Bukvic, Manuela Priolo, Alessandra Rosati

**Affiliations:** 1U.O.C. Medical and Laboratory Genetics, A.O.R.N., Cardarelli, 80131 Naples, Italy; marina.tarsitano@aocardarelli.it (M.T.); riviecciomaria@gmail.com (M.R.); mar.oro94@libero.it (M.O.); manuela.priolo@aocardarelli.it (M.P.); 2StressBioLab, Department of Medicine, Surgery and Dentistry “Schola Medica Salernitana”, University of Salerno, 84081 Baronissi, Italy; abasile@unisa.it (A.B.); arosati@unisa.it (A.R.); 3Department of Clinical and Experimental Medicine, University of Foggia, 71121 Foggia, Italy; nazzareno.capitanio@unifg.it; 4Medical Genetics Section, University Hospital Consortium Corporation Polyclinics of Bari, 70124 Bari, Italy; nenad.bukvic@policlinico.ba.it

**Keywords:** KRAS isoforms (KRAS4A; KRAS4B), sotorasib (AMG 510), p.G12C mutation

## Abstract

**Simple Summary:**

KRAS mutations have critical roles in the etiology of cancers such as NSCLC, CRC, and PDAC. However, therapeutic targeting of KRAS has proven difficult due to the lack of typical drug-binding sites. Recent improvements have seen the emergence of inhibitors specially tailored for the p.G12C mutation, with Sotorasib showing promising results in NSCLC. Nonetheless, the CodeBreaK 200 study found no significant difference in overall survival between Docetaxel and Sotorasib. This study compares the structural configurations of KRAS isoforms KRAS4A and KRAS4B and analyzes the effects of mutations on drug binding. The present study reveals unique aggregation propensities in wild-type and mutant isoforms, highlighting the complexities of KRAS as a therapeutic target. Sotorasib’s stable structure may allow for more effective binding to KRAS4B, despite the steric constraints imposed by KRAS4A and its mutations. This emphasizes the necessity for additional research into the complicated dynamics of KRAS targeting in cancer therapy.

**Abstract:**

Kirsten Rat Sarcoma Viral Oncogene Homolog (KRAS) gene variations are linked to the development of numerous cancers, including non-small cell lung cancer (NSCLC), colorectal cancer (CRC), and pancreatic ductal adenocarcinoma (PDAC). The lack of typical drug-binding sites has long hampered the discovery of therapeutic drugs targeting KRAS. Since “CodeBreaK 100” demonstrated Sotorasib’s early safety and efficacy and led to its approval, especially in the treatment of non-small cell lung cancer (NSCLC), the subsequent identification of specific inhibitors for the p.G12C mutation has offered hope. However, the CodeBreaK 200 study found no significant difference in overall survival (OS) between patients treated with Docetaxel and Sotorasib (AMG 510), adding another degree of complexity to this ongoing challenge. The current study compares the three-dimensional structures of the two major KRAS isoforms, KRAS4A and KRAS4B. It also investigates the probable structural changes caused by the three major mutations (p.G12C, p.G12D, and p.G12V) within Sotorasib’s pocket domain. The computational analysis demonstrates that the wild-type and mutant isoforms have distinct aggregation propensities, resulting in the creation of alternate oligomeric configurations. This study highlights the increased complexity of the biological issue of using KRAS as a therapeutic target. The present study stresses the need for a better understanding of the structural dynamics of KRAS and its mutations to design more effective therapeutic approaches. It also emphasizes the potential of computational approaches to shed light on the complicated molecular pathways that drive KRAS-mediated oncogenesis. This study adds to the ongoing efforts to address the therapeutic hurdles presented by KRAS in cancer treatment.

## 1. Introduction

Kirsten Rat Sarcoma Viral Oncogene Homolog (KRAS) is a significant driver in several cancers, including non-small cell lung cancer (NSCLC), colorectal cancer (CRC), and pancreatic duodenal adenocarcinoma (PDAC). Despite extensive studying of KRAS’s significance in many cancers, developing effective treatments has remained difficult. For decades, KRAS has been labeled as untreatable because of its absence of conventional drug-binding sites [1].

However, recent developments have opened possibilities for the creation of personalized inhibitors that precisely target the p.G12C mutation in NSCLC, raising the prospect of new therapeutic alternatives [2].

Notably, the release of Sotorasib (AMG 510), a KRAS p.G12C inhibitor, has sparked considerable interest because of its clinical efficacy in NSCLC patients. Despite these encouraging findings, the recent CodeBreaK 200 study, a clinical trial, found no statistically significant difference in overall survival (OS) between individuals treated with Docetaxel and Sotorasib. This unexpected result raises questions regarding the efficacy of KRAS inhibition as a monotherapy and necessitates further investigation into the intricate molecular and structural details underpinning KRAS isoforms and variants [3].

The KRAS gene is alternatively spliced to produce the two primary isoforms of the KRAS protein, KRAS4A (188 amino acids) and KRAS4B (189 amino acids), both of which are required for the protein’s function. The isoforms have the same G domains, which contain the first 165 amino acids. The last segment of the G domain is encoded by the 5′ fragment of exon 4 and comprises three differences at residues 151, 153, and 165 before the 23 aa HVR (hypervariable region). The G domain of the protein functions as both the catalytic and switching domain, binding to GDP/GTP and establishing linkages with effectors, exchange factors, and GTPase-activating proteins (GAPs) [4,5].

The two KRAS splice variants’ HVRs differ significantly and contain information that is critical for membrane localization. Three post-translational changes occur in the C-terminal CAAX sequence included in the HVR: farnesylation, CAAX proteolysis, and carboxyl methylation of the resultant C-terminal prenylcysteine [6]. With the exception of KRAS4B, every RAS protein has one or two palmitoylated cysteines right before the CAAX sequence. KRAS4B, on the other hand, has a polylysine sequence instead of a palmitoylation site, which allows it to engage electrostatically with the negatively charged inner layer of the plasma membrane [4].

KRAS4A is unique in that it has two short polybasic regions (PBRs) and is palmitoylated, making it a dual membrane-targeting motif. In the C-terminal membrane-targeting region, cysteine 180 undergoes palmitoylation, which separately promotes effective interaction with the plasma membrane. The PBRs and palmitoylation are both necessary for the finest possible signaling efficiency [7].

These post-translational modifications are crucial for KRAS4A and KRAS4B membrane-anchoring processes and predict differences in their kinetics for membrane attachment, which is required for signaling and, as a result, RAS function [8].

The polybasic region of KRAS4A is typically linked to signaling via the PI3K pathway, which is involved in the growth, proliferation, and survival of cells. On the other hand, KRAS4B primarily activates the MAPK pathway, which controls cell division and proliferation, due to its farnesylated CAAX box motif [9,10] (Figure 1).

The two isoforms of the KRAS protein exhibit distinct structural, functional, and distributional characteristics. KRAS4A is primarily found in specific cell types such as neurons, with a high prevalence in the brain. Despite its historical underrepresentation in scientific studies, recent research has underscored its significance in regulating neuronal pathways and influencing neuronal growth and functionality. This highlights the need for further exploration of KRAS4A’s role in neurobiology [4].

KRAS4B, on the other hand, has a more diverse distribution across numerous cell types and tissues, with higher amounts found in the liver, colon, lungs, and other organs. The diverse distribution of these isoforms suggests that they each play a unique role in different cellular and tissue contexts [4].

KRAS4A and KRAS4B may play specific roles in the tumor microenvironment and are essential for the development of tumors. The role of these isoforms in carcinogenesis has been explored in recent publications. KRAS4A expression improves the ability of tumor cells to adapt to demanding environments like hypoxia, in contrast to KRAS4B, which is present in both stem and progenitor cells. Based on the profiling of KRAS splice variants, recent studies have demonstrated differences in the expression levels of KRAS4A and KRAS4B in various tissues [11].

KRAS4A to KRAS4B expression ratios are influenced by two factors: the type of tumor under investigation (lung, pancreatic, or colorectal cancer) and whether the surrounding tissue is malignant or normal. KRAS4B mRNA, for instance, was expressed at higher levels than KRAS4A mRNA in patients with non-small cell lung cancer (NSCLC) [12]. However, KRAS4A expression was higher in melanoma and colon cancer cell lines, while the colon displayed similar levels of splice variants [7].

The most dynamically regulated RAS isoform, according to a recent study, is KRAS4A, which is upregulated in the stomach, intestine, kidney, and heart during preterm development [13].

Various in vivo and in vitro studies have highlighted the importance of understanding KRAS isoform-specific effects. For example, the response of KRAS p.G12C inhibitors like ARS-853 and ARS-1620 demonstrated varied effectiveness based on the GTPase activity of the isoforms, indicating potential resistance mechanisms linked to the specific KRAS variant [14].

In the case of Sotorasib, its efficacy on KRAS p.G12C mutation can be influenced by the presence of specific KRAS isoforms.

Interestingly, Sotorasib has shown different levels of effectiveness against the two primary isoforms of KRAS: KRAS p.G12C and NRAS p.G12C. While it effectively inhibits KRAS G12C, it is significantly more potent against NRAS G12C. Structural studies have revealed that this difference in potency is due to a single amino acid variation in the binding pocket of these isoforms (Histidine-95 in KRAS versus Leucine-95 in NRAS) [15,16].

This study aims to introduce an additional layer of complexity to the ongoing investigation by comparing the three-dimensional structures of KRAS4A and KRAS4B. The presented analysis suggests that alterations in the pocket domain of these isoforms could result in steric hindrance for Sotorasib. The atropisomeric structure of Sotorasib prevents rotation around a bond, typically a single bond, leading to the formation of stereoisomers that are stable and isolable at room temperature. This unique characteristic significantly contributes to its therapeutic efficacy and selectivity. By maintaining stable conformation, Sotorasib can bind more effectively to the KRAS4B pocket, thereby overcoming the steric challenges posed by the KRAS4A isoform and its mutants. In an additional aspect of this study, a computational analysis is conducted to investigate potential structural changes resulting from the three most common mutations: p.G12D (36%), p.G12V (23%), and p.G12C (14%). It is noteworthy that 80% of all oncogenic mutations in KRAS mutant tumors are located within codon 12 [17,18]. This analysis reveals distinct aggregation propensities of wild-type (Wt) and mutant isoforms, providing insights into the complexities of targeting KRAS as a therapeutic strategy.

## 2. Materials and Methods

### 2.1. Structural Analysis of KRAS Isoforms

A 3D model of the protein structures of KRAS4A and KRAS4B was created using Phyre2, a powerful bioinformatics tool. The target protein’s sequence is compared to a large database of known protein structures, structurally related proteins are found, and the information is then used to build a reliable 3D model of the target protein. Additionally, we used computational methods like Chem3D molecular modeling and simulation to analyze differences in their tertiary structures [19].

### 2.2. Analysis of KRAS Variants

Individual analyses were conducted on the three main KRAS variants, p.G12C, p.G12D, and p.G12V. To introduce these mutations into the KRAS4A and KRAS4B structures, we carried out 3D investigations. Chem3D computational software was used to predict and assess changes in protein structure as well as potential changes in functional areas.

### 2.3. Aggregation Propensity Analysis

To investigate the aggregation propensity of wild-type and mutant KRAS isoforms, we employed bioinformatics tools and molecular dynamics simulations using GalaxyWEB (https://galaxy.seoklab.org/cgi-bin/submit.cgi?type=HOMOMER, accessed on 18 March 2024 and https://galaxy.seoklab.org/cgi-bin/submit.cgi?type=HETEROMER, accessed on 18 March 2024). Chem3D computational software was used to predict and assess changes in protein structure [20,21].

## 3. Results

The computational structural analysis revealed subtle, yet significant, differences between the two main KRAS isoforms, KRAS4A and KRAS4B. Two of these differences were the accessibility of potential Sotorasib-binding sites and the location of significant functional domains.

A cryptic pocket on the surface of the target protein, KRAS p.G12C, can only be accessed by a structural motif of specific conformation. The cryptic pocket in the mutant protein p.G12C is formed by amino acids H95, Y96, and Q99, as well as the Switch II domain (AA 60–76) [22]. Sotorasib’s design includes an axially chiral component that is necessary for binding to the KRAS p.G12C pocket. This perfect match is critical to the molecule’s function because it enables Sotorasib to selectively inhibit the mutant protein, effectively blocking the downstream signaling pathways that drive cancer cell development. The formation of the axially chiral link in Sotorasib includes a significant rotational barrier, which is essential to its stability and function. This barrier is caused by energy differences imposed by steric strain and other contributing factors, which ensure that the atropisomers do not easily interconvert at physiological temperatures [23].

### 3.1. Analysis of KRAS4A and KRAS4B Changes in the Pocket Domain

A distinct structural difference in the pocket domain has been identified between the two isoforms, KRAS4A and KRAS4B. This comparison highlights a hidden pocket formed by the arrangement of amino acids H95, Y96, and Q99, depicted in a space-filling model, along with amino acid C12 (Figure 2, Panel A). A closer inspection reveals a significant disparity between the two isoforms, with amino acid Q99 in the KRAS4A isoform rotated by approximately 20°. Figure 2 offers a detailed view of the pocket domain, underlining that residue Q99 is located near S65, a crucial component of the Switch II domain. The green line in the KRAS4A isoform illustrates changing curvature, suggesting that the Sotorasib molecule must adapt its shape to fit properly within the pocket domain.

The atropisomeric conformation of Sotorasib is essential to maintain its linear structure and ensure proper drug binding that aligns with the geometry of the KRAS4B pocket. The steric constraints imposed by a conformational alteration in the KRAS4A isoform could potentially reduce Sotorasib’s inhibitory activity. As a result, Sotorasib may not bind and position itself optimally due to these structural dissimilarities in KRAS4A (Figure 2, Panel B).

### 3.2. Aggregation Propensity Analysis

KRAS-GTP interaction adopts a changed conformation in switches I and II of the G domain, after which KRAS is activated and binds to downstream molecules as a monomer or dimer to drive a variety of signaling cascades [24].

This section examines the propensity of KRAS4A and KRAS4B to generate homodimeric and heterodimeric complexes. While the wild-type isoforms appeared to form stable-only dimeric complexes, mutant isoforms showed significant alterations in their aggregation behavior. Specifically, GalaxyWEB tool molecular dynamics simulations revealed that the KRAS4A 12C dimer had a more compact structure than the KRAS4B 12C dimer. This enhanced structural tightness in KRAS4A p.G12C causes changes in the pocket domain, potentially preventing Sotorasib from entering it. Furthermore, KRAS4A p.G12C’s ability to form tetrameric complexes reduces Sotorasib’s binding ability. The four SWITCH II domains in this tetrameric structural configuration prevent the drug from binding to the appropriate pocket (Figure 3, Panel A).

In contrast, the KRAS4B p.G12C dimer maintains the pocket domain’s integrity, allowing Sotorasib to be correctly positioned (green line), demonstrating the molecule’s specificity to this domain, and it seems unable to form tetrameric structures. This study increases the evidence of Sotorasib’s specificity for the KRAS4B isoform (Figure 3, Panel B).

Additionally, all isoforms can form heterodimeric complexes involving the key residues H95, Y96, and Q99 in the interaction with the inhibitor. According to the research conducted using the Galaxyweb tool, it reduces the likelihood of drug interaction. The dislocation of these residues changes the spatial arrangement and conformation of key amino acids, thus influencing Sotorasib’s ability to bind the target site in both isoforms (Figure 3, Panels C and D).

The analysis was then extended to examine the conformational changes occurring in the pocket domain with the introduction of the p.G12D and p.G12V mutations into the KRAS4A and KRAS4B isoforms. The comparative analysis, illustrated in Appendix A, reveals that the Switch II domain (orange) approaches the cryptic pocket (blue), creating steric hindrance that inhibits Sotorasib interaction (green circle). This observation is consistent with the drug’s ineffectiveness against these two mutated forms.

Next, it was determined whether the p.G12D and p.G12V mutations could influence the creation of complex structures. Specifically, the KRAS4A isoform forms dimeric structures in the presence of the p.G12D and p.G12V variants, with the p.G12V variant increasing the chance of tetrameric structures. In contrast, the KRAS4B isoform shows an increased tendency to form both dimeric and tetrameric structures in the presence of p.G12D and p.G12V variants (Appendix A).

### 3.3. Exploring Druggable Sites: KRAS4A and KRAS4B Regions as Potential Drug Targets

Investigations into the structural dynamics of homo- and heterodimeric complexes have elucidated potential constraints associated with the covalent KRAS p.G12C inhibitor, Sotorasib. These insights are pivotal for the re-evaluation of a spectrum of investigational pharmacological agents. Such research may explicate observed phenomena such as a diminished binding affinity or adverse pharmacological profiles in certain compounds. For instance, MRTX849 (adagrasib) has demonstrated efficacy analogous to Sotorasib, augmented by an extended biological half-life and the capacity for central nervous system infiltration. Additional inhibitory molecules, including LY3537982, GDC-6036 (divarasib), and D-1553 (garsorasib), are undergoing various phases of clinical experimentation. Nevertheless, despite the auspicious outcomes in preclinical and preliminary clinical evaluations, impediments such as resistance development and deleterious effects remain, necessitating continual investigative efforts [3,25].

The examination of the three-dimensional configurations of homo- and heterodimeric entities may contribute valuable insights into the operational dynamics of the non-covalent pan-KRAS inhibitor, BI-2865. This inhibitor exhibits selective affinity for the quiescent conformation of KRAS, thereby conserving NRAS and HRAS integrity. It impedes nucleotide exchange, thereby obstructing the activation cascade of both the native KRAS protein and an extensive array of KRAS mutations, encompassing G12A/C/D/F/V/S, G13C/D, V14I, L19F, Q22K, D33E, Q61H, K117N, and A146V/T. This inhibition mechanism has been shown to effectively attenuate neoplastic proliferation in murine models [3].

Empirical research has demonstrated that the pan-KRAS inhibitor exhibits a pronounced affinity for the GDP-bound conformation of various KRAS isoforms, thereby interrupting the association with nucleotide exchange factors such as SOS1 and hampering their subsequent activation. Structural examination has ascertained that the inhibitor’s allosteric site remains conserved among diverse KRAS mutants. Moreover, amino acid alterations within KRAS, notably at positions H95, P121, and S122, are determinants of the inhibitor’s efficacy, modulating either susceptibility or resistance. These observations imply that the majority of KRAS mutants oscillate between their active and inactive states and are dependent on nucleotide exchange for their activation, rendering them susceptible to this novel category of inhibitors. Consequently, this presents a broad therapeutic option for patients with KRAS-driven cancers [26].

In this manuscript, a critical aspect based on structural evolutionary conservation across multiple isoforms and complex assemblies has been highlighted. A thorough comparison of both wild-type (Wt) and mutant versions of the KRAS4A and KRAS4B isoforms, as well as predicted complex dimeric and tetrameric configurations, revealed two prominent protruding regions. The first region, located between the p-loop and the Switch I domain, extends from amino acid 23 to 35. The second region is in the terminal part of the catalytic domain, between amino acids 116 and 126, which includes residues P121 and S122 that influence sensitivity or resistance to the pan-KRAS inhibitor. The identification of these accessible regions within the KRAS protein structure may herald significant expansion of the therapeutic target landscape and provide a greater range of pharmacological interventions aimed at inhibiting KRAS-driven cellular proliferation within tumor settings. Finally, this finding emphasizes the significance of structural integrity and heterogeneity within the KRAS protein family, which could have far-reaching implications for the development of targeted treatment modalities (Figure 4).

## 4. Discussion

KRAS mutations are linked to the development of several carcinomas, including non-small cell lung cancer (NSCLC), colorectal cancer (CRC), and pancreatic ductal adenocarcinoma (PDAC). Historically, the KRAS protein has been considered “undruggable” due to its lack of conventional drug-binding grooves, which has hampered the development of effective therapies despite substantial study into its oncogenic role.

Over the past decade, significant progress has been made in targeting RAS oncogenes, transitioning from once “undruggable” targets to the clinical approval of two drugs and the development of several more. The initial challenge stemmed from the absence of accessible binding sites on RAS proteins, especially KRAS p.G12C mutants. However, inhibitors like Sotorasib and adagrasib have successfully targeted these mutants, showing promising responses in KRAS p.G12C-mutant NSCLC. This led to their accelerated FDA approval in 2021 and 2022. Despite their efficacy, rapid drug resistance has emerged, highlighting the need for combination therapies [27]. The immunosuppressive tumor microenvironment caused by RAS mutations suggests potential synergy with immune checkpoint blockade (ICB) therapies, although translating this into clinical success has been challenging. Understanding these dynamics has spurred innovative strategies combining RAS inhibition with ICB, leading to progress in preclinical models and ongoing clinical trials. This journey reflects both successes and ongoing challenges in effectively targeting RAS mutations in cancer therapy [27].

Recent advancements in targeting the p.G12C mutation, especially in NSCLC, have fueled optimism for developing customized inhibitors. However, the CodeBreaK 200 study’s finding that there is no significant difference in overall survival between treatments with Docetaxel and Sotorasib underscores the complexities of effective KRAS neutralization. KRAS mutations are common in many cancers and are associated with significant changes in cellular metabolism, such as increased glucose uptake, enhanced glycolysis, and altered glutamine metabolism. These metabolic changes sustain tumor growth and create a hostile microenvironment that facilitates immune evasion. Studies highlight the impact of KRAS mutations on metabolic reprogramming, offering insights that could lead to novel therapeutic strategies targeting metabolic vulnerabilities in RAS-driven cancers. The RAS family proto-oncogenes (KRAS, NRAS, HRAS) play a crucial role in cellular signaling and cancer biology [28]. Oncogenic mutations lock RAS in an active state, promoting cell proliferation, suppressing apoptosis, and altering cellular metabolism and the tumor microenvironment. Despite RAS’s significant role in cancer, therapies targeting it are limited due to its complex regulation and mutation variability across cancer types [29].

For these reasons, combination therapies investigating the ability to target downstream metabolic pathways and restore metabolic homeostasis in cancer cells are becoming increasingly important for KRAS p.G12C inhibitors. Researchers have identified genes known as “collateral dependencies” (CDs) that become crucial when inhibiting KRAS p.G12C. These CDs are notable for their capacity to increase cellular susceptibility to treatment. By classifying these CDs into pathways that affect KRAS signaling and other cellular processes, it becomes feasible to develop combination therapies. These therapies, which target the CDs, work in synergy with KRAS p.G12C inhibition and show potential for boosting anticancer activity, both in vitro and in vivo [29]. Additionally, mechanisms behind the development of resistance to KRAS p.G12C inhibitors allow cancer cells to evade targeted therapies. Understanding these mechanisms is crucial for developing more effective treatments and overcoming drug resistance in cancer therapy. Research focused on NSCLC with KRAS driver mutations, in particular, the KRAS p.G12C mutation, shows that co-mutations in STK11 and/or KEAP1 make these cancers notably resistant to standard therapies, including targeted and immune-based treatments. While KRAS p.G12C inhibitors like adagrasib (MRTX-849) represent a significant advancement, resistance remains a challenge [30]. In this study, a structural comparison investigation of the KRAS4A and KRAS4B isoforms, as well as an examination of the potential structural variations caused by common mutations such as p.G12C, p.G12D, and p.G12V, has provided critical insights. Although the G domains of both isoforms are identical, their membrane-targeting sequences differ significantly. KRAS4B’s polylysine sequence and lack of palmitoylation promote its increased interaction with the plasma membrane. In contrast, KRAS4A has dual-membrane-targeting domains that are both polybasic and palmitoylated, which improves signaling potency and plasma membrane localization. These findings provide insight into the varied roles that KRAS variants play in carcinogenic signaling pathways by highlighting subtle variations in their subcellular trafficking processes.

The computational analyses presented here revealed discrete structural differences within the pocket domains of KRAS4A and KRAS4B, as well as the aggregation tendencies of both wild-type and mutant isoforms, shedding light on the complex challenges inherent in KRAS therapeutic targeting. This study emphasizes the importance of tailoring treatment techniques to specific isoforms, as changes in structural properties, such as the organization of crucial functional areas, might modify the binding suitability of therapeutic options, influencing their therapeutic efficacy.

Moreover, the description of the two proteins’ regions accessible to pharmacological interaction opens up promising opportunities for therapeutic intervention. These findings not only highlight isoform-specific differences, but also identify possible drug-binding sites within distinct domains. Understanding these complexities is critical for developing more effective therapeutic methods and moving forward in our effort to unravel the molecular complexities of KRAS isoform biology.

## 5. Conclusions

Despite the development of specific inhibitors, the KRAS protein remains a tough obstacle in oncological therapy. Our comprehensive comparative analyses of KRAS isoforms and variants have underlined the intricate structural and functional nuances of KRAS, emphasizing the importance of developing tailored therapeutic approaches that respond to specific isoforms and individual patient profiles to effectively combat KRAS-driven malignancies. The therapeutic issue is complicated by the various aggregation tendencies of KRAS isoforms, both wild-type and mutant. Future research should focus on understanding the processes regulating the change in aggregation behavior, which may open up new avenues for innovative therapeutic options in the fight against KRAS-mediated malignancies.

## Figures and Tables

**Figure 1 cancers-16-02389-f001:**
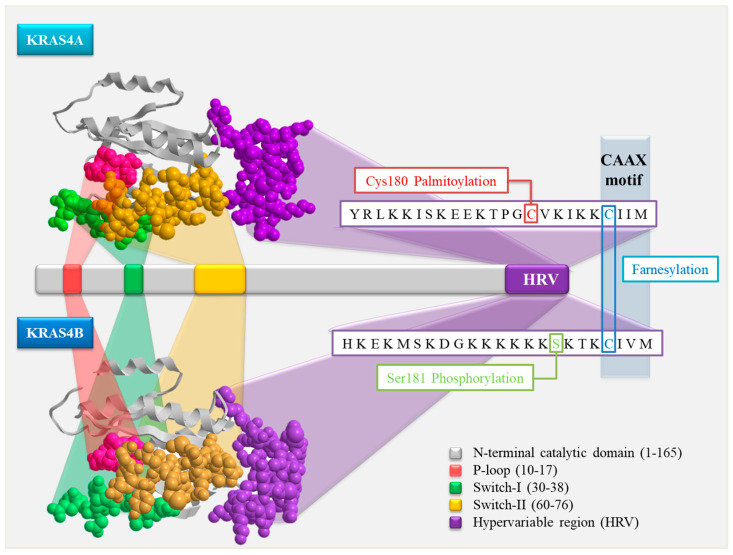
A schematic representation of the three-dimensional structures of the two KRAS isoforms (KRA4A and KRAS4B), with the domains P-loop (AA10–17), Switch I (AA30–38), and Switch II (AA60–76), and the unique amino acid sequence of the HRV domain marked in space-filling mode. The AA residues linked to farnesylation, phosporylation, and palmytoylation are marked in the amino acid sequence.

**Figure 2 cancers-16-02389-f002:**
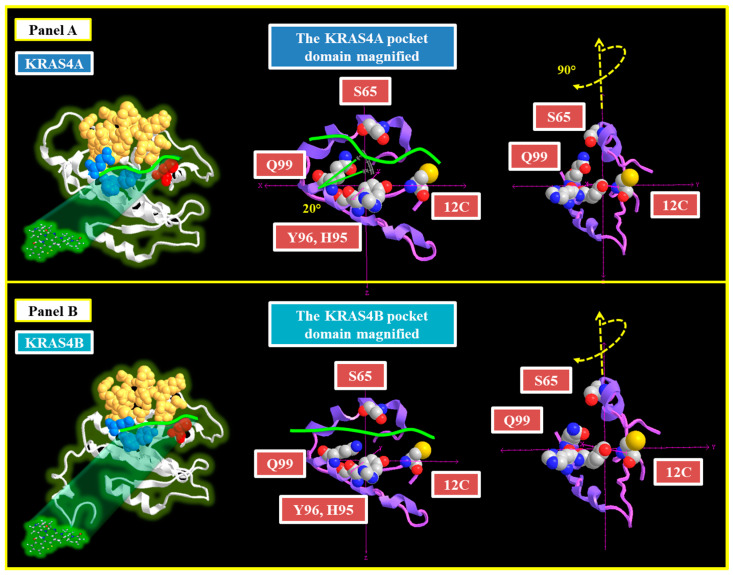
The figure shows the three-dimensional structures of the two isoforms KRAS4A and KRAS4B ((**Panel A**) and (**Panel B**), respectively). In both panels, the Switch II domain region (amino acids 60–76) is highlighted in orange, the amino acid residues H95, Y96, and Q99 are in blue, and the C12 mutation is in red. In the center, an enlargement of the pocket domain reveals a 20° rotation of residue Q99 toward residue S65 (**Panel A**). Additionally, on the left, the same pocket is shown rotated by 90°, providing a clearer view of the proximity between residue Q99 and residue S65. In **Panel B**, showing the KRAS4B structure, the same steric hindrance is not present, and residues S65 and Q99 appear further apart. The green line indicates the potential modification of Sotorasib needed for correct access to the pocket domain.

**Figure 3 cancers-16-02389-f003:**
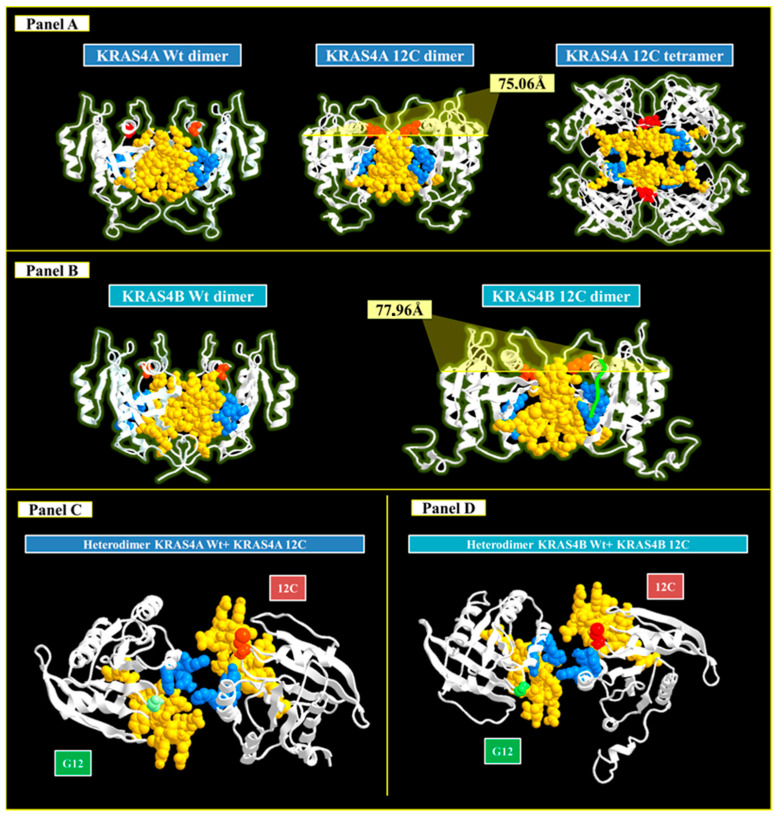
A 3D representation of the homodimeric structures formed by the KRAS4A (**Panel A**) and KRAS4B (**Panel B**) isoforms. Notably, both isoforms generate homodimeric structures, but they differ in form. The KRAS4A homodimer is more compact (75.6A) compared to the KRAS4B homodimer (77.96A), which could result in additional steric hindrance affecting the proper positioning of Sotorasib. **Panels C** and **D** also show the heterodimeric structures of KRAS4A and KRAS4B.

**Figure 4 cancers-16-02389-f004:**
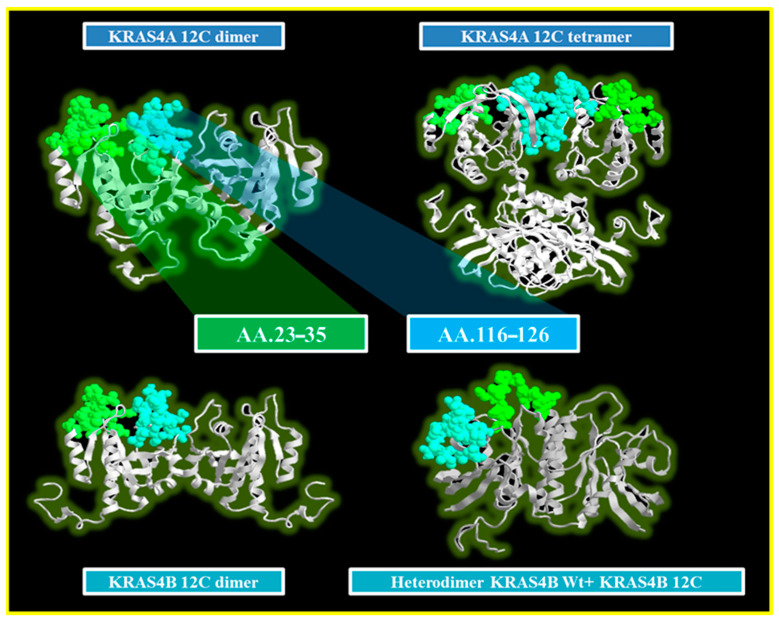
The image shows the predicted homo- and heterodimeric structures for KRAS4A and KRAS4B. In all structures, two protruding regions (AA23–35 in green and AA116–126 in light blue) are highlighted. These regions, based on their spatial location, could serve as potential binding sites for a drug capable of recognizing both isoforms in all possible conformations.

## Data Availability

The data that support the findings of this study are available on request from the corresponding author.

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
