# Peer review of "The Target Therapy Hyperbole: “KRAS (p.G12C)”—The Simplification of a Complex Biological Problem"

_cancers, 2024, doi:10.3390/cancers16132389_

Round 1
Reviewer 1 Report
Comments and Suggestions for Authors
An article by Dr. Chetta and a group elaborating on the structural configurations of KRAS isoforms KRAS4A 18 and KRAS4B analyzes the effects of mutations on drug binding. It's a unique way of explaining the complexity of KRAS with therapeutic implications. A few things must be addressed before it is ready for acceptance. They are as follows:
1. In line 248, where the authors mention the resistance development, authors must mention the recent work where the resistance landscape of KRAS G12C inhibitors is studied thoroughly ( PMID: 37729426 and PMID: 31138768).
2. Discussion should be further extended by mentioning the possible role of KRAS inhibitors in altered cancer metabolism (PMID: 33870211) and immune therapies (PMID: 38471457).
3. In Figure 1, use any dark color or fluorescent color instead of light green so that it will be more prominent.
Author Response
Reviewer 1
An article by Dr. Chetta and a group elaborating on the structural configurations of KRAS isoforms KRAS4A 18 and KRAS4B analyzes the effects of mutations on drug binding. It's a unique way of explaining the complexity of KRAS with therapeutic implications. A few things must be addressed before it is ready for acceptance. They are as follows:
- In line 248, where the authors mention the resistance development, authors must mention the recent work where the resistance landscape of KRAS G12C inhibitors is studied thoroughly (PMID: 37729426 and PMID: 31138768).
- Discussion should be further extended by mentioning the possible role of KRAS inhibitors in altered cancer metabolism (PMID: 33870211) and immune therapies (PMID: 38471457).
- In Figure 1, use any dark color or fluorescent color instead of light green so that it will be more prominent.
Response:
Thank you for your thorough review and constructive feedback. We appreciate your insights and recommendations for enhancing our manuscript. Below are our responses to your comments:
- Line 248 has been revised to mention recent research on the resistance landscape of KRAS G12C inhibitors, including PMIDs 37729426 and 31138768. We believe that this feature provides a more complete picture of resistance evolution.
2.The discussion section has been expanded to include the potential involvement of KRAS inhibitors in cancer metabolism and immunological treatments. To support these changes, we have added the suggested references (PMID: 33870211 and 38471457).
- We updated Figure 1 by replacing the light green color with a more noticeable color. The suggested change significantly improves the figure's visibility and clarity.
We hope these revisions address your concerns and enhance the quality of our manuscript.
Reviewer 2 Report
Comments and Suggestions for Authors
The authors analyzed the steric structure of KRAS4A and 4B and reported that Sotrasib may be more effective against KRAS4B.
This is a very interesting analysis. According to this report, analysis of isoforms, in addition to KRAS mutations, may be necessary to consider treatment options.
There are an increasing number of reports on the effects of different KRAS isoforms on cancer.
If the authors can find in vivo or in vitro reports showing that the effects of KRAS inhibitors such as Sotrasib differ depending on the KRAS isoform, please cite them.
Author Response
Reviewer 2
The authors analyzed the steric structure of KRAS4A and 4B and reported that Sotrasib may be more effective against KRAS4B.
This is a very interesting analysis. According to this report, analysis of isoforms, in addition to KRAS mutations, may be necessary to consider treatment options.
There are an increasing number of reports on the effects of different KRAS isoforms on cancer.
If the authors can find in vivo or in vitro reports showing that the effects of KRAS inhibitors such as Sotrasib differ depending on the KRAS isoform, please cite them.
Response:
Thank you for your positive feedback and insightful suggestions. We are grateful for your comments and have made the following revisions to our manuscript.
We have conducted a thorough literature review and identified several in vivo and in vitro studies that report the differential effects of KRAS inhibitors, such as Sotorasib, based on KRAS isoforms. These studies have been cited into introduction.
We believe these additions provide a more comprehensive analysis and support the need to consider isoform-specific effects in treatment planning.
Reviewer 3 Report
Comments and Suggestions for Authors
The authors provided an overview of Sotorasib (AMG 510), a KRASG12C inhibitor, and insights into its two major isoforms, 4A and 4B. As a premise, in their description of the efficacy and safety of Sotorasib, the authors primarily cited the "CodeBreaK 200 trial." This trial, conducted after the approval of Sotorasib, has several issues. Therefore, when discussing Sotorasib, the authors need to base their explanation on the Phase II "CodeBreaK 100 trial." The lack of difference in OS results in the "CodeBreaK 200 trial" can be attributed to several factors, including the allowance of crossover. The information that serves as the premise and background for the authors' research is incorrect. Moreover, I need help understanding the results of this paper and whether the KRAS isoforms truly affect the efficacy of Sotorasib. Additionally, the writing structure is complicated to read, suggesting a need for a comprehensive review of the overall composition.
Author Response
Reviewer 3
The authors provided an overview of Sotorasib (AMG 510), a KRASG12C inhibitor, and insights into its two major isoforms, 4A and 4B. As a premise, in their description of the efficacy and safety of Sotorasib, the authors primarily cited the "CodeBreaK 200 trial." This trial, conducted after the approval of Sotorasib, has several issues. Therefore, when discussing Sotorasib, the authors need to base their explanation on the Phase II "CodeBreaK 100 trial." The lack of difference in OS results in the "CodeBreaK 200 trial" can be attributed to several factors, including the allowance of crossover. The information that serves as the premise and background for the authors' research is incorrect. Moreover, I need help understanding the results of this paper and whether the KRAS isoforms truly affect the efficacy of Sotorasib. Additionally, the writing structure is complicated to read, suggesting a need for a comprehensive review of the overall composition.
Response:
Thank you for your detailed review and critical insights. We appreciate your comments, which have prompted us to make significant improvements to our manuscript. Below are our responses to your concerns:
Our intention is not to question the differences between the "CodeBreaK 200" and "CodeBreaK 100" trials for the KRAS G12C inhibitor Sotorasib, as we are aware that they differ significantly in design, objectives, and outcomes.
It is evident that "CodeBreaK 100" established the initial efficacy and safety of Sotorasib, leading to its approval. In contrast, "CodeBreaK 200" aimed to compare Sotorasib's performance against standard chemotherapy, demonstrating its advantages in progression-free survival (PFS) and safety, though it did not show an advantage in overall survival (OS) due to certain study design factors.
The mention of CodeBreaK 200 in the article relates to the ASCO Living Guideline, Version 2023.2, and the "Therapy for Stage IV Non–Small-Cell Lung Cancer With Driver Alterations." This guideline clearly states that while Sotorasib showed significant improvements in PFS and had a better safety profile, there was no statistically significant difference in OS between the two groups, likely due to the crossover allowed in the study design. Therefore, our study could help clarify not only the impact of crossover but also identify the specific patient settings where the drug can have a real effect.
Moreover, throughout the manuscript, we have revised the description of the results, aiming to enhance clarity and readability. We have also incorporated more detailed references to the figures.
Round 2
Reviewer 1 Report
Comments and Suggestions for Authors
All concerns addressed, and ready for acceptance.
Reviewer 3 Report
Comments and Suggestions for Authors
I couldn't understand the authors' arguments. I'm sorry.